# Supercritical Extraction Techniques for Obtaining Biologically Active Substances from a Variety of Plant Byproducts

**DOI:** 10.3390/foods13111713

**Published:** 2024-05-30

**Authors:** Filip Herzyk, Dorota Piłakowska-Pietras, Małgorzata Korzeniowska

**Affiliations:** 1Department of Functional Food Products Development, Faculty of Biotechnology and Food Sciences, University of Environmental and Life Sciences, 51-630 Wrocław, Poland; filip.herzyk@upwr.edu.pl; 2Wroclaw Technology Park, 54-413 Wrocław, Poland

**Keywords:** supercritical extraction, bioactive compounds, carbon dioxide, green extraction technique, byproducts

## Abstract

Supercritical fluid extraction (SFE) techniques have garnered significant attention as green and sustainable methods for obtaining biologically active substances from a diverse array of plant byproducts. This paper comprehensively reviews the use of supercritical fluid extraction (SFE) in obtaining bioactive compounds from various plant residues, including pomace, seeds, skins, and other agricultural byproducts. The main purpose of supercritical fluid extraction (SFE) is the selective isolation and recovery of compounds, such as polyphenols, essential oils, vitamins, and antioxidants, that have significant health-promoting properties. Using supercritical carbon dioxide as the solvent, supercritical fluid extraction (SFE) not only eliminates the need for hazardous organic solvents, e.g., ethanol, and methanol, but also protects heat-sensitive bioactive compounds. Moreover, this green extraction technique contributes to waste valorisation by converting plant byproducts into value-added extracts with potential applications in the food, pharmaceutical, and cosmetic industries. This review highlights the advantages of SFE, including its efficiency, eco-friendliness, and production of residue-free extracts, while discussing potential challenges and future prospects for the utilisation of SFE in obtaining biologically active substances from plant byproducts.

## 1. Introduction

The extraction of biologically active substances from diverse plant byproducts represents an overarching paradigm in the pursuit of sustainable and value-driven approaches to resource utilisation. Within this intricate landscape, supercritical extraction techniques have emerged as a pivotal avenue, rooted in a rich historical trajectory that mirrors the evolution of extraction methodologies. With origins dating back to the early 20th century, supercritical fluid extraction (SFE) has progressively evolved, driven by the burgeoning need for greener and more efficacious extraction processes. As the current paradigm shifts towards sustainable practices and waste valorisation, SFE stands as a linchpin, uniquely poised to encapsulate the twin objectives of enhancing extraction efficiency while adhering to eco-conscious principles. This article delves into the multifaceted domain of green extraction techniques, encapsulating the inherent attributes of supercritical extraction, its resolute alignment with environmental sustainability, and the pivotal role it plays in procuring natural extracts teeming with bioactive potential from plant byproducts.

## 2. Green Extraction Techniques

In recent years, there has been a growing interest in the development and adoption of green extraction techniques in the food industry, aiming to minimise environmental impact and promote sustainability [1]. According to Sagar et al. [2] these innovative methods encompass a variety of approaches, such as supercritical fluid extraction (SFE), enzyme-assisted extraction (EAE), microwave-assisted extraction (MAE), pulse electric field (PEF), ultrasound-assisted extraction (UAE), high-voltage electrical discharge (HVED), and pressurised liquid extraction (PLE). Green extraction techniques stand out for their reduced use of hazardous solvents, lower energy consumption, and higher selectivity in extracting valuable bioactive compounds. One particular area of interest lies in the extraction of compounds classified as generally recognised as safe (GRAS) by regulatory authorities, including essential oils, polyphenols, vitamins, and other functional ingredients. These GRAS compounds are known for their potential health benefits and are extensively used as flavourings, antioxidants, and natural preservatives in the food industry. Green extraction techniques offer a promising pathway for obtaining these compounds from various plant materials, food byproducts, and natural sources, leading to the development of safer, more sustainable, and health-promoting food products.

## 3. Main Purpose of Supercritical Extraction by Carbon Dioxide

The primary objective underlying supercritical extraction pertains to the acquisition of extracts characterised by elevated quality attributes, characterised notably by augmented purity and heightened concentration of the designated target compounds. This technique engenders a discerning capacity for the extraction of specific constituents of interest, thereby effecting the deliberate separation of desired components from extraneous substances, and concomitantly conferring the prospect of refining the resultant extract’s compositional fidelity. Additionally, supercritical extraction is considered a more sustainable and environmentally friendly method compared to solvent-based extractions, as supercritical CO_2_ is non-toxic, non-flammable, and easily recyclable [3,4].

It is imperative to underscore that the precise objectives and application scope of supercritical extraction are subject to variation contingent upon the specific industry and the nature of the target compounds under consideration. The technique inherently embodies a remarkable degree of versatility and adaptability, facilitating tailored extraction processes that cater to the unique demands inherent to each distinct application. This adaptability encompasses the capability to extract an expansive array of compounds, thereby accommodating the diverse and dynamic requirements that govern disparate applications within the broader context of supercritical extraction [5,6]

The supercritical extraction technique finds pervasive application across multifarious industries encompassing pharmaceuticals, food, cosmetics, and beverages [7]. Its utility is notably harnessed for the extraction of bioactive compounds derived from a diverse array of botanical specimens. This technique serves as a principal conduit for extracting essential oils from botanical matrices [8,9,10], concomitant with eliminating undesirable constituents that might impede subsequent processing endeavours, such as deleterious impurities or malodorous nuances. Moreover, supercritical extraction is pivotal in procuring aromatic and chromatic constituents from natural reservoirs [11], thereby underscoring its multifaceted and indispensable contributions to diverse industrial domains.

## 4. Enhance the Supercritical Extraction Efficiency Using Co-Solvents

The incorporation of co-solvents within the framework of supercritical fluid extraction (SFE) engenders the potential for augmented extraction efficacy and discriminatory capacity pertaining to specific bioactive moieties that inherently elude extraction through the medium of carbon dioxide [6]. These co-solvents, when judiciously introduced into the supercritical fluid matrix, predominantly comprising carbon dioxide (CO_2_), serve as modulating agents, orchestrating a refinement in solvent potency while concomitantly effectuating an enhancement in the solubilisation of specific compounds, notably those evincing elevated polarity profiles [4]. Typically existing in a liquid state, co-solvents are deliberately chosen to confer miscibility with supercritical CO_2_ under the precise nexus of operational pressure and temperature conditions. Among the pantheon of co-solvents frequently harnessed within SFE, ethanol, methanol, and water, emerge as salient examples, each poised to impart a distinctive impetus to the extraction process through their cooperative interactions with supercritical CO_2_ [12].

The utilisation of co-solvents confers an array of advantageous attributes. Notably, co-solvents engender heightened solubility for polar entities, a characteristic particularly pertinent due to the inherent solubility constraints manifested by supercritical carbon dioxide vis-à-vis polar molecules, exemplified by certain polyphenolic compounds or water-soluble vitamins. The judicious incorporation of compounds such as ethanol, methanol, or acetone [3], which exhibit propensities for hosting polar moieties, substantiates the augmentation of process selectivity [5]. Tailoring specific co-solvents to distinct compounds amplifies this selective propensity, thereby attuning the process for optimal yield [13]. Furthermore, co-solvents exercise a dual-faceted influence by facilitating the reduction of process temperature and pressure, thereby concomitantly heightening extraction efficiency.

However, the confluence of benefits proffered by co-solvents necessitates judicious and meticulous orchestration. A calibrated equilibrium must be struck in terms of co-solvent selection and concentration, as excessive co-solvent deployment bears the potential of inducing undue swelling in plant material or yielding undesired alterations in extract composition [14]. Moreover, the overall safety and suitability of co-solvents demand exacting scrutiny. Rigorous adherence to co-solvent criteria, inclusive of regulatory approval for food-grade applications and the imperative of evading residual compounds in the final extract, merits conscientious attention. Fundamental to ensuring the fidelity and safety of the extracted bioactive entities is the diligent orchestration of process controls and validation mechanisms within the context of co-solvent integrated supercritical fluid extraction paradigms [4].

## 5. Advantages and Disadvantages of the Supercritical Extraction Technique

Supercritical extraction using carbon dioxide (CO_2_) bestows a multitude of distinct advantages, albeit accompanied by certain limitations when contrasted with conventional extraction methodologies. The salient prowess of supercritical extraction lies in its intrinsic capacity for heightened selectivity, a feature that engenders the isolation of target compounds while assiduously precluding the co-extraction of undesirable constituents within the resultant extract. This discerning selectivity assumes a mutable character, amenable to manipulation through the modulation of extraction parameters such as pressure and temperature during the process [15]. This dynamism facilitates the attainment of elevated purity benchmarks in the ultimate extract.

An additional hallmark of supercritical fluid extraction (SFE) resides in its amenability to operation under relatively moderate temperatures, thereby conferring a pivotal benefit of preserving the structural integrity and biological activity of compounds sensitive to thermal degradation [16]. This facet becomes conspicuously advantageous when contemplating the extraction of bioactive entities from natural sources, wherein the propensity for molecular degradation necessitates temperate conservation strategies. Evidencing its environmental soundness, carbon dioxide has garnered recognition from regulatory entities such as the FDA as an eco-friendly, non-toxic, and non-flammable agent, thereby eclipsing conventional solvents employed in extraction [17]. The extracts procured via supercritical CO_2_ are designated solvent-free, rendering them amenable to utilisation across pharmaceutical, food, and cosmetic sectors. Furthermore, the facile removal of CO_2_ through depressurisation obviates the presence of residual solvents in the final product, affirming the extract’s purity [18].

The resultant attributes intrinsic to supercritical fluids have garnered extensive utilisation across an array of matrices originating from both botanical sources [16,19,20], encompassing pomace [3,4,7,16,21], oils [17], herbs and spices [8,9], as well as crustacean byproducts [13]. This inherent adaptability affirms its applicability across a diverse spectrum of industries, thereby concertising its status as an inherently versatile technique. Notwithstanding the manifold advantages underpinning the purity and qualitative attributes of resultant extracts, supercritical extraction remains a technology replete with certain inherent challenges. Foremost among these are the substantial initial investment costs requisite for the establishment of high-pressure infrastructure and specialised apparatus. This financial burden augments the overall implementation costs of the technique within industrial contexts. Additionally, the seamless operationalisation of supercritical extraction mandates a nuanced grasp of the intricate interplay of extraction parameters, thereby demanding specialised expertise and experience for their adept optimisation [12,22].

Moreover, the scalability of supercritical extraction, vis-à-vis conventional extraction methodologies, bears inherent complexities. These complexities stem from the intricate orchestration of large-scale extraction systems to maintain both control and sustained extraction throughput. This endeavour engenders logistical and operational quandaries that underscore the challenges of expansion [13].

Notably, the energy-intensive nature of the extraction system emerges as a salient drawback. This high-energy consumption is requisite to engender the requisite supercritical conditions, thereby accentuating the energy demands of the process. Concomitantly, the efficiency of supercritical extraction vis-à-vis compounds evincing elevated polarity exhibits a discernible limitation. This constraint, however, can be surmounted through the judicious integration of co-solvents to enhance solubility and extend the extraction scope to encompass high-polarity compounds [12,20].

## 6. Possible Modifications of the Supercritical Extraction

Modifications to supercritical extraction methodologies engender heightened prospects for augmenting extraction efficiency, selectivity, and precision in the acquisition of desired compounds from source materials. A focal point of these refinements revolves around the adept selection of suitable co-solvents, instrumental in effecting the extraction of specific compounds, particularly those endowed with polar characteristics or elevated molecular weights. The judicious orchestration of optimal operating pressure and process temperature serves as an avenue for bolstering extraction yields or facilitating the precise targeting of compounds of interest. Concomitantly, the integration of cyclodextrins, and cyclic oligosaccharides, imparts a dimension of augmentation to compound extraction and stability enhancement [22].

To this extent, preparatory techniques, such as grinding, particle size reduction, and drying, are strategically administered prior to SFE to amplify the surface area, thereby amplifying extraction efficiency. Sequential extraction, entailing multifarious extraction stages under diverse conditions, emerges as a strategic protocol to orchestrate the selective extraction of distinct compound categories. Concomitant applications of fractionation and separation methodologies within the SFE paradigm are harnessed to effectuate the isolation of specific compounds or the expurgation of undesired constituents [23].

Moreover, SFE’s interface with other extraction methodologies, including microwave-assisted extraction (MAE) and ultrasound-assisted extraction (UAE), bespeaks a synergistic symbiosis, underpinning augmented yields or enhanced specificity for target compounds. It warrants acknowledgement that the efficacy of the supercritical extraction process hinges upon a profound comprehension of the intrinsic attributes of the target compounds, their solubility profiles, and the desired extraction outcomes. This prescient cognizance underscores the integral imperative of informed decision-making and process customisation within the domain of supercritical extraction [24].

## 7. Other Methods of Extraction

In addition to the supercritical extraction described in this review, various methods listed below are used to obtain bioactive compounds.

### 7.1. Classical (Conventional) Methods of Extraction (CSE)

Within the gamut of traditional extraction methodologies, each distinguished by their inherent merits and tailored suitability to specific applications, stand out techniques such as maceration and Soxhlet extraction. Maceration entails the immersion of source material within a solvent, affording a streamlined procedure characterised by its adaptability to a wide spectrum of compounds, spanning both polar and non-polar domains. However, it is noteworthy that this technique bears the drawback of prolonged extraction periods and the substantial utilisation of solvents [25].

Concurrently, the Soxhlet extraction, facilitated through specialised equipment, facilitates solvent recirculation through the matrix, a process optimised for the extraction of semi-polar and non-polar entities. Nevertheless, the technique is marred by environmental concerns, protracted operational timelines, and prodigious solvent consumption [5].

### 7.2. Steam Distillation and Hydrodistillation

Steam distillation and hydrodistillation are used for the extraction of essential oils and crude pigments from aromatic botanical materials. This entails the passage of water vapour or water through the plant substrate, transporting volatile compounds that subsequently condense and segregate from the aqueous phase. While efficacious in capturing volatile constituents, this approach is less amenable to compounds susceptible to thermal degradation and beset by lengthy temporal requirements [26].

### 7.3. Solvent Extraction

Solvent extraction, by contrast, harnesses organic solvents to engender compound extraction from botanical specimens. Renowned for its versatility, this method spans a diverse array of compounds, with subsequent solvent removal and extract concentration facilitated via a rotary evaporator. Nonetheless, this procedure is beset by temporal constraints and is less attuned to thermolabile species [13].

### 7.4. Liquid–Liquid Extraction (LLE)

Liquid–liquid extraction (LLE), an apt selection for liquid matrices, circumvents phenolic compound degradation through ambient temperature operation. Notwithstanding its utility, this technique is constrained by the exigency of perilous and costly chemicals, protracted analysis timelines, and escalated rates of degradation due to external and internal factors [2].

### 7.5. Solid-Phase Extraction (SPE)

Solid-phase extraction (SPE), alternatively, garners acclaim for its ergonomic utility and accelerated throughput, dwarfing its liquid–liquid counterpart. Its reproducibility is a hallmark virtue, albeit the method is encumbered by financial demands and its applicability is restricted when confronted with volatile analytes, susceptible to evaporation losses [27].

## 8. Other Green Extraction Techniques

Green extraction methodologies, poised as sustainable alternatives, underscore a paramount commitment to ecological harmony while ameliorating the environmental footprint. Notably, microwave-assisted extraction (MAE) assumes prominence, a dynamic approach heralded for its curtailed solvent utilisation, diminished energy outlays, and catalytic augmentation of microwave-driven extraction kinetics. However, its merits are juxtaposed with certain constraints, notably encompassing equipment-related capital expenditures and suboptimal applicability concerning non-polar and thermolabile compounds [13,24].

### 8.1. Ultrasound-Assisted Extraction (UAE)

Ultrasound-assisted extraction (UAE) surfaces as a commendable facet, wherein ultrasonic waves incite fervent agitation within the extraction milieu, efficaciously accentuating mass transfer mechanisms and consequent extraction efficiency amplification. This culminates in elevated product yield, abbreviated processing intervals, and abated chemical and energy profligacy. This method’s efficacy, while intriguing, rests upon judicious optimisation of ultrasound frequency and system geometry to engender maximal operational efficacy, reflecting the nuanced landscape that underpins its potential [28].

### 8.2. Enzyme-Assisted Extraction (EAE)

Embarking upon enzyme-assisted extraction (EAE), a paradigm underscored by its environmentally concordant attributes and the utilisation of water as the extraction solvent, emerges as a potent choice, especially for compounds necessitating rapid extraction. However, the high-cost quotient affiliated with enzymes for voluminous sample sizes, coupled with nuanced factors including sample moisture, particle dimensions, hydrolysis duration, enzyme concentration, and compositional considerations, bears pertinence as pivotal determinants of efficacy [22].

### 8.3. Pressurised Liquid Extraction (PLE)

Pressurised liquid extraction (PLE), manifesting as a propitious avenue for solid sample isolation of biomolecules, reverberates with virtues encompassing temporal parsimony and reduced solvent expenditure. Nevertheless, the heightened equipment outlay coupled with its incompatibility with samples bearing low levels of target analytes refracts the discernment of its scope [12].

### 8.4. High-Voltage Electric Discharge (HVED)

High-voltage electric discharge (HVED), in a testament to energy frugality, capitalises on efficient biomolecule extraction. This unfolds within a condensed timeframe, paralleled by judicious solvent utilisation. The associated drawback lies within the compromised selectivity, attesting to the intricacy of its implementation dynamics [27].

### 8.5. Pulsed Electric Field (PEF)

A kindred technique, pulsed electric field (PEF), transpires as a transformative approach delineated by its divergence from conventional extraction paradigms, thereby underscoring the imperative of environmental stewardship. It not only truncates energy requisites but also abbreviates extraction timelines while concurrently elevating efficiency benchmarks. However, cognisance of the parameter selection congruent with each stage constitutes a quintessential imperative in achieving tangible efficacy [29].

## 9. Novel Green Extraction Techniques

The progressive landscape of extraction methodologies is perpetually underpinned by the development of innovative approaches aimed at surmounting the limitations intrinsic to conventional techniques. This evolutionary endeavour is entrenched in the quest to enhance efficiency, resilience, and the discriminatory prowess underlying the acquisition of invaluable compounds from diverse sources.

### 9.1. Supercritical Water Extraction (SWE)

Supercritical water extraction (SWE) and its complement, subcritical water extraction (SCWE), epitomise this trend. These techniques harness subcritical water conditions—both above and below the critical point—as a medium to concurrently conserve energy, curtail solvent consumption, and effectuate the extraction of compounds susceptible to thermal degradation [23,30].

### 9.2. The Instant Controlled Pressure Drop (DIC; Détente Instantanée Contrôlée)

The instant controlled pressure drop (DIC; Détente Instantanée Contrôlée) method, distinguished by its rapid pressure perturbation, catalyses the fracturing of plant cell walls, facilitating the liberation of intracellular constituents. This process obviates thermal degradation and solvent extravagance while concurrently engendering expeditious and efficacious extraction. Particularly salient in the context of heat-sensitive bioactive substances, DIC emerges as a transformative approach [31].

### 9.3. Supercritical Liquid Chromatography (SFC)

Supercritical liquid chromatography (SFC), an emerging analytical separative paradigm, leverages supercritical fluids as mobile phases, a dynamic trajectory garnering prominence owing to its high-resolution attributes and ecological compatibility [32].

### 9.4. Pressurised Hot Water Extraction (PHWE)

Pressurised hot water extraction (PHWE) pivots on the application of pressurised hot water, fusing the capacity to extract bioactive compounds with the pursuit of a greener alternative to traditional solvent-based extraction strategies [27].

As these innovative extraction modalities burgeon, a panorama of possibilities unfolds. The trajectory of technological advancement and concurrent scientific exploration portends the potential for these techniques to catalyse a revolutionary transformation within the extraction realm, ultimately underpinning the proliferation of suitable and efficacious practices across diverse industries.

## 10. Safety of Solvents Using Green Extraction Techniques

As alluded to earlier, the utilisation of carbon dioxide (CO_2_) as the solvent within the framework of supercritical fluid extraction (SFE) enjoys a preeminent reputation for its generally recognised as safe (GRAS) status. This accolade is attributed to its inherent non-toxic and non-flammable nature, ecological benignity, and complete volatilisation upon extraction. The unique amalgamation of gaseous and liquid traits exhibited by supercritical CO_2_ facilitates the efficacious extraction of target compounds from diverse plant substrates, unmarred by residual contaminations or olfactory nuances, thus engendering an unblemished profile of purity and safety of the ultimate extract. The exceptional safety and environmental affinity attributed to CO_2_-based SFE have engendered its ascendancy as a green and sustainable alternative to conventional solvent-based extraction methodologies. It is incumbent upon the practitioner to diligently adhere to robust process controls, equipment upkeep, and validation protocols to safeguard both safety and quality parameters. However, the inherently benign nature of CO_2_ endows SFE with considerable promise as a cogent contender for the extraction of bioactive compounds. This is particularly noteworthy in the context of the escalating demand for products that are not only safer and sustainable but also uphold stringent standards of quality across multifarious industries.

## 11. Natural Extract Obtained from Plant Byproducts

In recent years, the increased emphasis on sustainability and resource optimisation has resulted in a shift in the perception of food industry waste (Figure 1). These once-discarded residues, which include agricultural byproducts, food processing residues, and discarded entrees, are increasingly appreciated for their untapped nutritional value. Such waste materials contain a wealth of bioactive compounds, including phytochemicals, antioxidants, fibre, and proteins, which provide potential health benefits. The exploitation of these compounds through modern extraction techniques has emerged as a promising route to transform food industry waste into value-added resources. Historically relegated to uses such as animal feed or destined for landfill, these waste streams are now lucrative raw materials for the production of bioactive extracts. Strategic extraction of bioactive compounds from these discarded substrates not only reduces the burden of waste disposal but also offers a sustainable way to extract valuable functional ingredients [26]. This paradigm shift not only contributes to waste reduction but also highlights the potential to revolutionise the food industry’s approach to resource utilisation, aligning with today’s desire for a closed-loop economy and environmentally friendly practices [3].

The residual components often encompass botanical fractions that are not directly harnessed for their primary utility yet retain significant bioactive constituents. The proper extraction and utilisation of these constituents serve as a catalyst for waste reduction and the generation of value-added commodities within industrial contexts. Pomace extracts, originating from the residual solids of fruit juicing processes, exemplify this trend, heralding applications as pigments, flavour enhancers, and nutraceutical agents, enriched with polyphenols, anthocyanins, and assorted antioxidants. Akin in relevance is the utilisation of pomace residues originating from oilseed processing, rendering abundant stores of fatty acids and antioxidants. These byproducts constitute indispensable commodities across the food, cosmetic, and pharmaceutical sectors [33]. Notably, the residue ensuing from sugar cane juice extraction, specifically sugar cane fibre, emerges as a reservoir of antioxidants, underpinning applications spanning biomaterials encompassing biofuels and construction materials [2]. Similarly, the peel of citrus fruits unfolds as an abundant repository of essential oils and flavonoids, harbouring a diverse array of applications spanning perfumery, cosmetics, and cleansing agents [34].

The appropriate integration of these natural extracts augments the paradigm of waste reduction and engenders sustainable practices. This paradigm shift underscores the inexorable trajectory towards value creation within a multiplicity of industries, a manifestation of the harmonisation between ecological responsibility and industrial innovation [28].

## 12. Bioactive Compounds Obtained Using Supercritical Fluid Extraction from Byproducts

Leveraging the capabilities of supercritical fluid extraction (SFE), diverse arrays of bioactive moieties harbouring salutary health attributes can be meticulously extracted from byproducts. This paradigmatic shift has garnered pronounced attention from both industrial and scientific domains, foregrounding the exploration of polyphenolic entities, encompassing flavonoids, phenolic acids, and stilbenes. Remarkable for their robust antioxidant potency, these polyphenols orchestrate the quelling of free radical entities, consequently mitigating the propensity for cardiovascular maladies. Echoing their polyphenolic counterparts, carotenoids emerge as constituents of interest, manifesting antioxidant attributes that reinforce immunological resilience [6].

Intriguingly, the pursuit of alkaloids surfaces prominently within this ambit, with notable representation in the pharmaceutical arena as stimulants and analgesics. This category enlists notable molecules such as caffeine, nicotine, and morphine [25]. The roster of extractable entities extends further to encompass terpenes and terpenoids, distinguished aromatic compounds encapsulated within essential oils, their multifaceted biological activities are punctuated by their contribution to botanical flavour and aroma profiles [8,9]. Likewise, phytosterols, bearing structural semblance to cholesterol, engender cholesterol absorption reduction, thereby conferring cardiovascular benefits.

Saponins, lauded for their trifold attributes of anti-inflammatory, antioxidant, and immune-modulating properties, stand as integral constituents of this assortment [35]. Essential oils, revered for their antimicrobial, antifungal, and antioxidant dimensions, represent another noteworthy facet [25]. Furthermore, the significance of dietary fibres cannot be understated, given their pivotal role in bolstering digestive wellness and fostering satiety [36]. It warrants acknowledgement that the composite assortment and concentration of bioactive entities exhibit considerable variance across diverse plant species and specific botanical components.

In essence, the assimilation of an assorted spectrum of bioactive compounds into the human diet emerges as a cornerstone of nutritional vitality. This strategic dietary diversification engenders a holistic approach to wellness, encompassing multifarious physiological benefits that collectively conduce to overall health and well-being.

## 13. Comparison of the Effectiveness of Supercritical Fluid Extraction with Other Methods

Core parameters such as extraction time, temperature, pressure, solvent selection, and process-specific conditions are critical determinants of the efficiency and efficiency of extraction processes [37]. The proper management of these parameters is critical to optimise the final yield.

Extraction time has a direct influence on energy consumption and applicability of the technique, with operating time and temperature playing a major role, especially for thermolabile compounds. Among alternative processes, microwave-assisted extraction (MAE) (about 1 to 4 min) and ultrasound-assisted extraction (UAE) have significantly shorter extraction times compared to pressurised liquid extraction (PLE) (5 to 480 min) and supercritical fluid extraction (SFE) (13 min to more than 60 min). SFE requires the longest extraction time, which is its primary disadvantage [23].

MAE and PLE typically demand higher temperatures, which can facilitate the release of greater amounts of phenolic compounds; however, these temperatures must be limited due to the thermal instability of many compounds, most of which are unstable above 100 °C. PLE in particular performs at the highest temperatures, sometimes reaching as high as 225 °C [12].

For the preservation of thermolabile compounds, extraction methods operating at medium temperatures are preferred. SFE meets these requirements as the process temperature usually does not exceed 40 °C, which is closely related to the properties of carbon dioxide. In addition, the solvent used in SFE is easily removed from the final extract by changing the pressure. However, phenolic compounds are polar, which requires the use of co-solvents [13].

Organic solvents with higher polarity are generally more effective than non-polar solvents. In SFE, the use of solvent additives during extraction significantly increases the extraction efficiency of phenolic compounds, which is otherwise generally low using carbon dioxide alone. When conducted in an acidic environment, both PLE and UAE show significantly increased yields due to increased mass transfer through the cell walls. Furthermore, deep eutectic solvents (DES) in UAE also increase extraction efficiency [22].

Extraction efficiency and phenolic content are strongly dependent on the source material, cultivar and maturity stage, making it difficult to directly compare the performance of different extraction techniques relying solely on these factors. The literature indicates that MAE shows the highest total phenolic content (TPC), followed by PLE, UAE, and SFE. SFE, even when using a co-solvent, shows the lowest TPC, thus making it more suitable for the extraction of non-polar compounds and essential oils [37].

Therefore, the appropriate extraction process selection must take into account the extraction parameters and the properties of the raw material. The combination of different alternative methods can maximise the extraction efficiency and the yield of phenolic compounds. Among the alternatives, the combination of UAE with other methods appears to be the most promising. UAE in combination with PLE reduces extraction time, while UAE in combination with SFE increases antioxidant yields. This combined approach offers a versatile strategy for generating a variety of high-quality products from the same raw material. In particular, the combination of PLE and SFE allows the simultaneous extraction of non-polar compounds (via SFE) followed by polar phenolic compounds (via PLE) [10].

## 14. Application of Supercritical Fluid Extraction in Industry

The use of supercritical fluid extraction (SFE), particularly supercritical carbon dioxide (SC-CO₂), is gaining importance in the food and beverage industry due to its efficiency, environmental sustainability, and ability to produce high-purity extracts. SC-CO₂ is mainly used to extract essential oils and aromas from a variety of plant materials, including herbs, spices, citrus peels, and flowers [37]. These extracts are an integral part of flavourings in beverages, baked goods, savoury products, and confectionery [38]. The ability of this technique to preserve delicate flavour compounds makes it particularly valuable for extracting high-quality natural flavours. In addition, SFE is used to decaffeinate coffee and tea, taking advantage of its high selectivity to effectively remove the caffeine while preserving the essential flavours and aromas. This decaffeination method is favoured for its safety, environmental friendliness, and excellent ability to maintain product quality, making it an attractive and suitable choice for the industry [37].

Another significant industrial application of supercritical fluid extraction (SFE) is the extraction of bioactive compounds such as polyphenols, flavonoids, carotenoids, and sterols, mainly from waste and byproducts. Extracts are intended for inclusion in functional foods, dietary supplements, and nutraceuticals [23]. For the case in point, SFE is used to extract resveratrol from grape pomace and lycopene from tomato skins [2].

In addition, the extraction of natural pigments from plant materials by supercritical extraction has gained popularity in the food industry, driven by increasing consumer demand for clean-label products free from artificial additives. The technique is used to effectively extract anthocyanins from berries [15] and carotenoids from carrots and peppers, providing natural pigments for food products. The supercritical fluid phenomenon is also applied to extract antioxidants, such as tocopherols and polyphenols, from various plant sources, including green tea leaves and soybean oil [23]. These antioxidants are used to increase the nutritional value of food products and extend their shelf life by maintaining food quality and stability.

## 15. Further Perspective of Development

The trajectory of supercritical fluid extraction (SFE) is poised for further evolution through synergistic engagement with cutting-edge advancements. The ongoing refinement and diversification of extraction parameters, encompassing pressure, temperature, and solvent composition, hold promise for enhanced selectivity and efficiency [6]. The amalgamation of SFE with complementary techniques, such as co-solvent utilisation or novel sorbent materials, augments the versatility of the methodology, accommodating a broader spectrum of bioactive compounds. Moreover, the integration of process modelling and simulation platforms enables predictive insights into extraction dynamics, paving the way for precision-based process optimisation [10]. Concurrently, the convergence of SFE with emerging green solvents aligns the technique with sustainable principles, mitigating the environmental footprint and reducing reliance on conventional solvents. Leveraging advancements in automation and sensor technologies, real-time monitoring, and adaptive control of SFE processes could be realised, conferring unprecedented operational accuracy [6]. Furthermore, the quest for innovative SFE applications beyond conventional realms, including the extraction of rare compounds or tailored bioactive compound combinations, stimulates pioneering research horizons. The dynamic interplay of these perspectives coalesces to amplify the potential and propel the future evolution of supercritical fluid extraction as a preeminent tool in the domain of bioactive compound isolation and valorisation.

Withal, recent studies by Blejan et al. [39], Villamil-Galindo et al. [40], and Dróżdż et al. [41] underscore a burgeoning interest in utilising byproducts as ingredients possessing notable health-promoting attributes for human consumption. These materials are garnering attention from researchers due to their substantial concentration of antioxidants, notably polyphenols and anthocyanins, as well as their favourable lipid profiles. Additionally, they serve as viable natural colourant sources for industrial applications [42], rendering them promising candidates for augmenting functional food formulations over the long term [43]. This aligns with the core tenets of the circular economy, advocating for the efficient utilisation of resources and the adoption of contemporary processing methodologies. Byproducts encompass not only pomace resulting from juice or oil extraction but also various other plant morphological components, such as leaves (e.g., black currant), husks, stems, peels, and seeds, as specified by Staszowska-Karkut et al. [44], Blejan et al. [39], and Wójciak et al. [45], which exhibit substantial polyphenol and fibre content. Leveraging such byproducts represents a reservoir of valuable nutrients, recognised as a cost-effective resource offering manifold health benefits for both consumers and producers, fostering an environmentally sustainable and economically viable food chain [46]. Furthermore, utilising byproducts presents an alternative strategy for mitigating enzymatic browning, thereby extending product shelf life without compromising its sensory properties [40].

## 16. Conclusions

While the supercritical extraction process exhibits considerable potential in harnessing botanical source materials, along with their associated byproducts, its realisation necessitates fine-tuning to cater to bespoke exigencies. Predominantly, the scientific endeavour converges on augmenting extraction efficiency while minimising co-solvent deployment, concomitant with streamlining the extraction procedure to ensure the acquisition of extracts marked by optimal purity.

The purview of this extraction paradigm converges keenly on the byproducts emanating from processes involving juice and oil extraction, encapsulating its primary focal point.

## Figures and Tables

**Figure 1 foods-13-01713-f001:**
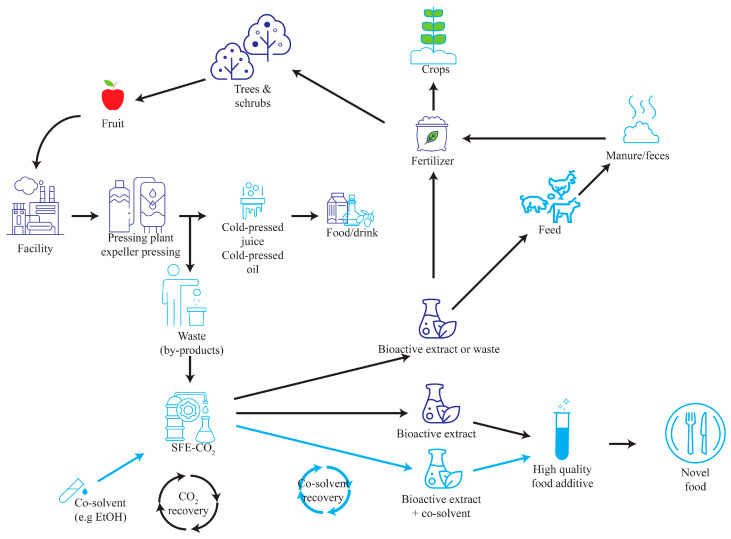
Possibilities of using byproducts extracted by supercritical extraction fluid.

## Data Availability

No new data were created or analysed in this study. Data sharing is not applicable to this article.

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
