# Peer review of "Supercritical Extraction Techniques for Obtaining Biologically Active Substances from a Variety of Plant Byproducts"

_foods, 2024, doi:10.3390/foods13111713_

Round 1
Reviewer 1 Report
Comments and Suggestions for Authors
This paper summarizes the principle of SFE and its advantages and application prospects in the utilization of by-product resources. However, the following problems are worthy of considering:
1. For SFE, the paper emphasizes supercritical carbon dioxide technology, and it is suggested that the topic should be specific for the whole content;
2. The content of the whole article is a bit scattered, so it is suggested that the article should not involve other extraction technologies too much, especially non-green extraction technology, so as to make the article more focused;
3. It is suggested to supplement the current industrial application status of supercritical technology, preferably with specific data support.
Reviewer 2 Report
Comments and Suggestions for Authors
Dear authors,
The paper “Supercritical extraction techniques obtaining biologically active substances from variety plant by-product” presents an overview of techniques applied for extraction of biological compounds from plants.
The paper contains valuable information but the author should address some aspects in order to accept the paper for publication:
1. English revision. Please pay more attention to English grammar and expression (the title should be: Supercritical extraction techniques for obtaining…).
2. While it is intended to be a comprehensive review the paper could be improved by adding tables summarizing the advantages and disadvantages of the mentioned green techniques.
3. A presentation of the information regarding the efficiency of SFE for extracting some specific by-products compared with other techniques would be useful.
4. There are other supercritical fluids used for SFE and a table presenting main characteristics and applications would definitely increase the value of the current paper.

Comments on the Quality of English LanguagePlease pay more attention to English grammar and expression (the title should be: Supercritical extraction techniques for obtaining…).
Round 2
Reviewer 1 Report
Comments and Suggestions for Authors
The authors have made careful revisions as required.